# The Effect of Erythritol on the Physicochemical Properties of Reformulated, High-Protein, and Sugar-Free Macarons Produced from Whey Protein Isolate Intended for Diabetics, Athletes, and Physically Active People

**DOI:** 10.3390/foods12071547

**Published:** 2023-04-06

**Authors:** Maciej Nastaj, Bartosz G. Sołowiej, Konrad Terpiłowski, Wiesław Kucia, Igor B. Tomasevic, Salvador Peréz-Huertas

**Affiliations:** 1Department of Dairy Technology and Functional Foods, Faculty of Food Sciences and Biotechnology, University of Life Sciences in Lublin, Skromna 8, 20-704 Lublin, Poland; 2Department of Physical Chemistry-Interfacial Phenomena, Maria Curie Skłodowska University, M. Curie Skłodowska Sq. 3, 20-031 Lublin, Poland; 3Wiesław Kucia’s Artistic School in Lublin, Wojciechowska 3, 20-704 Lublin, Poland; 4DIL German Institute of Food Technologies, Prof.-v.-Klitzing-Str. 7, 49610 Quakenbrueck, Germany; 5Department of Chemical Engineering, University of Granada, Avenida de la Fuente Nueva 12 S/N, 18071 Granada, Spain

**Keywords:** whey protein isolate, erythritol, aerated foods, macarons, rheology, texture, surface properties, Turbiscan, sportsmen nutrition

## Abstract

This study reports the possibility of obtaining sugar-free WPI-based macarons with erythritol addition. The whey protein isolate (WPI) solution (20%, *w*/*v*) was whipped, and erythritol was added to the foam at concentrations of 20, 40, and 60 g, with 125 g of almond flour. The rheological properties (τ, G′, G″, and tan (δ)) and stability of the macaron batters before baking were evaluated. In order to produce the macarons, the batters were solidified at 147 °C for 12 min. The textural and surface properties (roughness and color), as well as the microstructures and water activities, were determined for the macarons. It was feasible to produce macarons over the entire range of the tested erythritol content. Even the smallest amount of erythritol (20 g) facilitated the preservation of the macaron structure. The medium erythritol concentration (40 g) improved the stability of the batters and their rheology and was the most effective for air pocket stabilization during baking; however, its largest addition (60 g) resulted in an increase in the final macaron volume. The increased erythritol addition improved mechanical properties and shelf life, producing a smoothing effect on the macaron surfaces and having a significant effect on their color co-ordinates.

## 1. Introduction

French macarons are confectionery products that have attracted considerable interest from consumers in recent years. They are a very rich, meringue-based confection composed of beaten egg whites, granulated sugar, confectioners’ sugar, and almond flour. Macarons do not contain baking powder, and their aerated structure is obtained during baking by the release of steam from the beaten egg whites.

Saccharose is an essential component in providing macaron shells, and it also contributes not only to the flavor and sweetness but also provides their unique structure and lightness [1]. Sugar stabilizes the batter by increasing the viscosity of the water in its structure, making it denser and less likely to collapse when folded with other ingredients [2]. Although, various macaron recipes differ in their saccharose ratios, they are still considered very high [3]. 

Nowadays, the world has to face a serious issue of an increase in noncommunicable civilization diseases, and excessive sugar consumption is one of the most common factors to trigger this problem. At present, sucrose intake varies from 15% to 25% of the daily energy intake in many European Union countries, and being overweight as well as obese are common health sector concerns that are reported globally [4,5]. The introduction of a sugar tax on sweet goods is one of the governmental actions taken in European countries. Many of them expressed their great concern for public health and recommended the food industry reduce the sugar concentration in their products by 20% by the end of 2020 [6,7]. Therefore, appropriate measures should be taken to address these challenges and expand the range of reduced-sugar or sugar-free products on the market. In order to overcome the obesity problems, the food industries are being encouraged to reformulate food products by reducing the content of free sugars that increase body weight or adiposity. As confectionery products have grown in popularity, there must have been actions taken by the food industry and baking households to provide that [5,8,9,10,11].

As one of the polyols, erythritol is a very good substitute for traditional sucrose. The development of new modes of production via green and biotechnological techniques makes it increasingly cost-effective to obtain [12,13]. Erythritol sweetness is 60% of that of saccharose, and it does not trigger insulin secretion from the pancreas due to its very small caloricity (0.2 kcal g^−1^ against 4 kcal g^−1^ for sucrose). It has a very similar structure, organoleptic, and physical properties to sugar, making it widely used in sugar-free reformulated confectionery products [9,10]. Erythritol is not introduced to the circulatory system and is excreted out of the body completely intact [14,15]. When used in excess, some consumers have reported discomfort due to nausea, dizziness, diarrhea, abdominal pain, and bloating [16]. However, according to Storey et al. [17], an acceptable daily dose of 35–40 g does not cause any irritation in healthy volunteers. Additionally, the research on erythritol reports its superior properties over other polyols, like xylitol and sorbitol, for dental caries reduction [18].

The utilization of egg albumen for macaron production can cause a variety of problems related to salmonella in eggs [19], allergenicity [20], and the overbeating effect [21]. Egg albumin is a mixture of individual proteins with different molecular weights, charges, and isoelectric points, which is significant for foam formation [22]. Additionally, the issue of aging egg whites in macaron production is very energy and time-consuming as the water evaporation from the egg whites can take from 24 h up to 5 days in refrigeration temperatures, which is not the case for the reconstituted solutions of whey protein preparations.

Whey proteins demonstrate exceptional nutritional properties that exceed the biological value of egg albumin by 15%. For these reasons, and considering their unique functional properties, whey proteins are a desired and valued additive in the creation of high-protein functional foods intended for athletes and physically active people [23,24,25]. Additionally, with regard to whey proteins, the following health-promoting aspects deserve special mention: antioxidant properties [26], supporting cancer treatment [27] and patients receiving chemotherapy [28], cardiovascular diseases [29], immune system boost [30], and the management of diabetes, as whey proteins added to certain foods can reduce the glycemic index [31]. According to Boscasni et al. [32], whey proteins demonstrate beneficial effects on human metabolism, gut microbiota, and mental condition. In addition to the health motives, the use of whey protein preparations has an environmental justification, as the liquid whey, a byproduct of cheese production, can be fully utilized using separation techniques. Furthermore, new prospects for sourcing raw materials for this technology are very promising. Novel separation techniques have been and are being developed to produce high-protein whey preparations in a cost-effective manner. This means that the membrane technologies require less energy to operate, resulting in higher yields [33,34].

Considering the lack of reports in the literature on erythritol as a prospective saccharose substitute in confectionery products and the importance of whey proteins, it is essential to find out whether high-protein, sugar-free, and WPI-based macarons with acceptable properties can be produced for diabetics, athletes, and physically active people with increased protein requirements while reducing caloric value at the same time. This study provides information on the effects of the composition of macarons with the addition of WPI and erythritol, as these could be new potential compounds in the confectionery industry to produce novel functional foods. 

A special emphasis will be put on the surface properties of the macarons (roughness and color), as the research shows these affect the parameters of the final product and determine consumer behavior. Surface roughness is absolutely noteworthy as it relates to controlling smoothness, improving appearance, and limiting the defects (cracks) of the obtained product [35]. This parameter is still of particular significance for the palatability of food products [36], and in the case of confectionery products, it is important for their detachment from baking surfaces [37] and consumption time [9]. Therefore, in this paper, surface techniques will be highlighted to deliver extensive insights into the applications of whey protein isolate and erythritol in the confectionery sector. In this study, we also report on the rheological properties and the stability of macaron batters, as well as the texture, microstructure, and water activity of the finished products. 

## 2. Materials and Methods

### 2.1. Materials

The whey protein isolate (WPI) was kindly delivered, courtesy of Spółdzielcza Mleczarnia Spomlek (Radzyń Podlaski, Poland). The total protein content (91.4%) was calculated by the Kjeldahl method, determining the protein as N × 6.38. The powdered erythritol (99.5% purity) was purchased from PPH Stanlab (Lublin, Poland). Finely ground almond flour (52% fat, 20% protein, 13% fiber, 7.6% carbohydrates, 6% water, and 0.04% ashes) was obtained from Sempre Group Sp. z.o.o. (Pruszków, Poland). Their compositions were consistent with the manufacturers’ declarations.

### 2.2. Ingredient Quantity Criteria 

Four macaron reformulations were studied. The corresponding macaron compositions with the whey protein isolate and erythritol addition were determined by preliminary testing. Regarding the whey protein content, we were mostly guided by the average amount of complete protein that can be metabolized by an athlete in one meal (~30 g). When it comes to erythritol, we decided not to exceed the possible daily dose (35–40 g) per serving in any sample of the produced macarons batch. Table 1 presents the composition of the obtained macarons. For a comparison based on the traditional recipe, the control batch of macarons without erythritol was produced.

### 2.3. Preparation of the Macarons 

A WPI solution of 20% (calculated based on the pure protein content in the preparation, *m*/*v*) was obtained by hydration using the Heidolph MR 3002S magnetic stirrer (Schwabach, Germany) for 1 h. For equilibration, they were stored overnight at 7 °C. Before foam formation, the solutions were brought to an ambient temperature and mixed using the magnetic stirrer again for 1 h.

A total of 100 mL of the WPI solution was whipped to stiff peaks using the KitchenAid mixer (KitchenAid, St. Joseph, MI, USA) at speed 10 (280 RPM) for 5 min. Subsequently, the powdered erythritol was added as a dry ingredient into the wet foam at 0, 20, 40, and 60 g, and whipping continued for 3 min. Next, the almond flour was blended into (by spatula) each treatment at the rate of 125 g. 

The macaron batter of each batch was gently transferred to the piping bag and subsequently deposited onto the silicone baking pads (dimensions 39.2 cm × 28.6 cm) and solidified in the iCombiPro stove (Rational Selfcooking Center, Warsaw, Poland) at 147 °C for 12 min. This resulted in a total of 4 macaron batches for further investigations, which were ready to be analyzed almost immediately after manufacturing. Depending on the formulation, the obtained macarons were roughly 3.00–5.00 cm in diameter and 0.5–0.75 cm in height. Prior to the analyses, the macaron samples were kept in the desiccator. 

### 2.4. Rheology of the Macaron Batters before Solidification

The rheology of the batters relates to their stability and determines the further surficial, textural, and structural properties of the obtained macarons. All rheological analyses of the batters were conducted at 20 °C using the Kinexus Lab+ rheometer (Malvern Panalytical Ltd., Malvern, United Kingdom), employing plate-plate geometry. Two serrated plates with a 30 mm diameter were utilized to minimize the batter slippage, and the gap between them was 2 mm. The yield stress (τ) was measured at the steady shear rate of 0.01 s^−1^ for 2 min, and the maximum torque response allowed us to compute the yield stress. τ was identified as a certain stress value when the batter structure was disrupted. Oscillatory experiments were also conducted. The frequency sweep test was performed in the range of 0.1–10.0 Hz at 0.01% strain, and the changes in the storage (G′) and loss (G″) moduli and the phase angle values (σ) were measured. For comparison, the aforementioned parameters were recorded at a frequency of 1 Hz. Every rheological test was performed in triplicate immediately after each batter sample was produced. 

### 2.5. Batter Stability 

In addition to the rheological properties of the batters, the assessment of their stabilities was also of significant importance. Turbiscan analysis allows for monitoring all instability mechanisms (gravitational drainage, coalescence, and disproportionation) that occur simultaneously in the batters. This will facilitate the quality control of the finished product through the appropriate selection of the ingredients and the elimination of defects. Before solidification, the stability of the batters was examined in the Turbiscan Lab^Expert^ with a TLab Cooler cooling module (Formulaction, Smart Scientific Analysis, Toulouse, France) according to the method described by Nastaj et al. [24]. Turbiscan stability index (TSI) values were also determined using Turbiscan Easy Soft (Formulaction, Smart Scientific Analysis, Toulouse, France, www.formulaction.com/en/turbiscan-stability-index, accessed on 8 January 2023) based on the equation
(1)$$TSI=\sqrt{\frac{{\sum_{i=1}^{n}(x_{i}-x_{BS})}^{2}}{n-1}}$$
where x_i_ is the mean backscattering every 1 min of measurement, x_BS_ is the mean x_i_, and n is the number of scans.

### 2.6. The Texture of the Macarons

The following textural parameters of the macarons (hardness and fracturability) were measured by a TA-XT2i texture analyzer (Stable Microsystems, Godalming, England) equipped with a cylinder probe (70 mm diameter). The samples were compressed at a 1 mm s^−1^ crosshead speed and 50% deformation. Six measurements were taken per batch. The results were processed using the Texture Expert 1.22 software.

### 2.7. The Water Activity (a_w_) 

The water activity (a_w_) of the obtained macarons was determined by means of a water activity-meter LabMaster-aw (Novasina AG, Lachen, Switzerland) at 25 °C. The apparatus was calibrated for this dedicated temperature with the solutions of various salts according to the recommendations of the device manufacturer. This specific temperature will be optimal for the proper storage of the resulting macarons. The samples of the macarons were ground in a De Longhi KG200 home coffee grinder (De Longhi, Treviso, Italy) and placed into the plastic circular 35 mm-diameter capsules. Subsequently, the capsules were introduced into the chamber, and the analyzer lid was carefully closed. Each macaron sample was measured in triplicate.

### 2.8. Surface Properties of the Obtained Macarons (Roughness, Color Coordinates and Microstructure)

The surface of the obtained macarons was observed using an optical profilometer GT Contour Surface Metrology (Veeco, Tucson, AZ, USA). The surface roughness was calculated using the Vision64 software (Veeco). Color parameters, namely L*: brightness, a*: ±red-green, and b*: yellow-blue, were determined using a computer vision system (CVS); the method was invented and described earlier by Tomasevic et al. [38]. The microstructure was observed using a polarizing optical microscope Eclipse E600Pol (Nikon, Tokyo, Japan) at 40× magnification.

### 2.9. Statistical Analysis

The data were statistically analyzed by Statistica 13.0 (StatSoft, Cracow, Poland). All physicochemical analyses except for optical microscopy and batter stability in Turbiscan were performed at least in triplicate, with the arithmetic mean calculated. A one-way (different erythritol content) ANOVA was carried out, and the significant differences between the means were determined by Tukey’s posthoc test at *p* < 0.05. 

## 3. Results and Discussion

### 3.1. Rheological Properties of Macaron Batters before Baking

Table 2 shows the effect of erythritol on the yield stress (Pa), storage modulus (Pa), loss modulus (Pa), and phase angle (°) values of the obtained macaron batters before solidification. The rheological properties of the batters were influenced by erythritol addition. Increasing the erythritol content up to 40 g improved the rheological properties of macaron batters, and the E40 sample was the most rigid batter, which is reflected in the τ, G′, and G″ values. The E40 batter sample demonstrated the highest τ value (40.75 ± 2.1 Pa). It can be postulated that lower erythritol concentrations (20 g and 40 g) modified the continuous phase viscosity in the batter, providing a thicker lamellar film and a protein-specific effect on the interfacial elasticity of the systems. A less viscous medium allows for more air to be incorporated, resulting in a larger average bubble size [39]. This mechanism was earlier described by Nastaj et al. [9], who studied the rheological properties of similar systems and WPI-foams with erythritol that were designed to produce high-protein meringues. In the whole tested erythritol range, the most drastic decrease in *τ* was observed for the E60 sample (21.30 ± 1.1 Pa). The decrement in τ at the highest concentration of erythritol can be associated with the increase in batter viscosity that prevented the proteins from participating in the formation of the interfacial film. Nastaj et al. [35] and Luck et al. [40] explained that higher saccharose concentrations decreased the rheology of WPI foams. Sucrose creates less favorable thermodynamic conditions for whey protein unfolding and the development of protein—protein interactions. Thus, more proteins can be involved in film formation and stabilization due to reduced protein aggregation in the presence of sugar [41,42]. From the similarities in the chemical structure of erythritol (to sucrose), it can be assumed that the same mechanism occurs here as well.

The lowest G′ and G″ values were observed for the E60 sample, which confirmed the decrement in the rheological properties. The improvement in the rheological properties of the E20 and E40 samples is further confirmed by the phase angle values, which were 14.05 ± 0.05 and 13.98 ± 0.02, respectively. The differences in G′ and G″ and the domination of the elastic component (G′) over the viscous one (G″) observed for all samples indicate the interactions between the whey proteins and erythritol in the batters. Such behavior is also characteristic of weak physical gels [43]. For batters, foams, or gels that demonstrate relatively low yield stress values, greater energy input leads to sample damage, which will be confirmed by G″ decreases. For foams or weak gels exhibiting generally low τ values, a higher energy input can result in sample damage, which will lead to G″ decreases. Madadlou et al. [44] studied WPI gels with erythritol addition and observed decreases in the values of G’ and G″, hypothesizing that this polyol can increase the hydration of the proteins, reducing the evolution of the noncovalent bonds between the particles and impairing the development of the gel. Additionally, Cai et al. [45] stated that the erythritol particles could interfere with the formation of the gel protein network. 

For the aerated systems, the δ decrease is associated with the more elastic characteristics of the macaron batters. It should also be noted that, for the batters with an erythritol concentration of up to 40 g, with improved rheological properties, their phase angle values corresponded to the yield stress values. The lower δ was, the more elastic the macaron batter was, so it was more susceptible to elastic deformation [9,21].

### 3.2. Batter Stability

Figure 1 presents the evolution of the transmission and backscattering spectra and the Turbiscan stability index (TSI) values of the macaron batters before solidification. The occurrence of the following phenomena in the macaron batters could be identified: drainage, represented by a peak in the backscattering, and collapse, indicated by a peak in the transmission profile. The most stable sample was the E40 sample, which was reflected in the distribution of the scans. It is also worth noting that batter collapse was spectacularly diminished. The samples E0 and E60 exhibited totally different profiles that proved the smallest stabilities of the obtained batters. As described by Nastaj et al. [35] and Martínez-Padilla et al. [46], the increment in transmission showed that the batters were becoming more transparent, proving the occurrence of ongoing destabilization processes. Both the drainage and coalescence phenomena resulted in an increase in the liquid phase, so there was an increment in transmission and a decrease in backscattering. Disproportionation in relation to a smaller number of larger bubbles in the macaron batters resulted in an increase in transmission [46]. Larger erythritol additions led to increased viscosity in the food systems. According to Nastaj et al. [35] and Yang and Foegeding [47], viscosity is a key parameter in preventing air bubble coalescence in the batters. Hao et al. [48] reported larger drainage in the foam obtained from the liquid eggs and erythritol. Martínez-Padilla et al. [46] claimed that Oswald ripening (manifested by an increase in air bubble size over time) underwent air diffusion, which was caused by the pressure difference between the adjacent air bubbles. The distribution of the scans for the E20 and E40 batter samples showed that this phenomenon could be reduced when erythritol was added. Yang and Foegeding [49], who studied the effect of sucrose on the stability of WPI-based foams, claimed that their stabilities were improved due to retarded drainage. Based on the similarity of saccharose to erythritol, it can be assumed that an analogous phenomenon occurred for erythritol as well.

TSI is a very useful parameter for monitoring the destabilization kinetics of the macaron batters, and its value varies from 0 to 100. According to the TSI methodology, the values of TSI that are larger than 10 are reserved for very unstable food systems, like macaron batters. The E0 and E60 samples demonstrated the smallest stability. The increase in erythritol up to 40 g in the batter led to systematic TSI decrements. The E40 sample demonstrated the largest stability, which was confirmed by the lowest TSI value. The Turbiscan analyses validated the earlier observations when the rheological assessment of the corresponding macaron batters was made.

### 3.3. Surface Properties of Macarons

It is very important to emphasize that the thermal solidification of macaron batters, even without the erythritol addition, was possible, which is a very promising prospect for some consumers, especially in terms of having a low tolerance to erythritol. The macarons obtained over the entire range of the tested erythritol exhibited no quality defects, such as ruptures or cracks, which are very common for confectionery products like meringues, angel food cakes, and sponge cakes.

Figure 2 shows the optical profilometer images, and Table 3 presents the roughness parameters and color co-ordinates of the obtained macarons. The surface roughness was largely correlated with the erythritol content. The roughness parameters showed that the increasing erythritol concentrations promoted the gradual smoothing effect, with the obtained macarons exhibiting lower surface roughness, which could be clearly viewed in the profilometer images. The E60 macaron sample was the smoothest (R_a_ = 17.71 ± 2.97 μm), and the roughest structure was represented by the control sample without erythritol (R_a_ = 37.51 ± 2.49 μm). The analogous smoothing effect was described by Nastaj et al. [9] and Nastaj et al. [35], who produced and analyzed high-protein meringues from WPI and erythritol as well as from WPI and saccharose. A similar observation was made by Arunepanlop et al. [50], who reported that sponge cakes could exhibit various surfaces (from rougher to finer) with different sugar concentrations, and this was directly related to protein film rupture and air bubble collapse [47]. For the roughness and microstructure of the macarons, the vitrification phenomenon described by Mensink et al. [51] remained essential and allowed us to explain the differences in the surface structures when the amorphous, glassy matrix was produced by the sugars around the proteins. Berry et al. [52] pointed out the significance of continuous bubble growth in the batter affecting different surface properties during baking. The observed tendency with the roughness of the macarons could be associated with the higher viscosity of the batters because increasing erythritol concentrations delayed the movement of air bubbles up to the macaron surface during solidification. Based on the chemical similarity of sucrose to erythritol, it can be postulated that an analogous phenomenon also occurred for erythritol. Madadlou et al. [44] emphasized that interparticle, electrostatic, and hydrophobic interactions could alter the surface properties of food systems containing whey proteins and polyols. The studies by Díaz-Ramírez et al. [53] reported that whey proteins compete with sucrose for water; therefore, as the concentration of proteins increases, the stability of the sucrose decreases, which causes it to crystallize when subjected to heat, and the resulting crystals might have an impact on the surficial and textural properties of the final product.

It can be observed that the composition of the macarons exhibited a significant effect on their color co-ordinates (Table 3). Increasing the concentration of erythritol in the macarons had a major impact on the tested color parameters. Larger erythritol additions produced surfaces with higher luminescence (L) and lowered a* and b* co-ordinates. In practice, for a* and b*, this meant that the obtained macarons were characterized by a redder (or less green) and more yellow (less blue) color. According to Psimouli and Oreopoulou [54], the color of traditional confectionery goods containing saccharose was a result of Maillard browning and the caramelization of saccharose. However, erythritol has no accessible aldehyde group and is unable to get involved in the Maillard reaction [55,56]. This explained why the confectionery products with erythritol were lighter and demonstrated lower consumer acceptability. However, in this case, it should not be considered a disadvantage as it opens up the possibility for the use of food dyes. The color tinting of macarons is a very common confectionery practice, and higher L values with increasing erythritol concentration will make the macaron batter absorb the tint more easily. When analyzing the surficial data, the following dependency could be noted between the surface roughness and the luminescence of the macarons: as the roughness decreased, the luminescence increased. The reason for the increasing luminescence at lower roughness values was probably the consequence of lower light scattering with increasingly more regular and smoother surfaces [57]. It is worth noting that the color measurement of confectionery products (by common methods) is hardly representative; therefore, the application of a computer vision system should be recognized as a superior alternative to the traditional method for measuring the color of macarons.

### 3.4. Texture and Water Activity of Macarons

Table 4 shows the effect of erythritol concentration on the textural parameters (hardness and fracturability) and water activity of the obtained macarons. The macarons without erythritol (E0) demonstrated the largest hardness, and this parameter decreased as the erythritol concentration increased. The largest erythritol addition (E60) produced macarons with the least hard structure. Based on the textural properties, it can be noted that the values of hardness and fracturability were relatively similar. For all macaron samples, the decrease in hardness was associated with a decrease in fracturability. This mechanism was observed and explained earlier by Nastaj et al. [9], who obtained and analyzed the mechanical properties of high-protein meringues with the addition of whey proteins and erythritol. Raikos et al. [58] postulated that protein gelation upon heating was crucial in the confectioneries, and it determined the texture of the final products. Wijayanti, Bansal, and Deeth [59] claimed that the addition of polyols to the native protein solutions was reported to delay heat-induced protein aggregation and gel formation. Polyols can cause a delay in the development of a protein network and cause a weaker structure due to less freely available water, which results in a soft texture that is typical of sweet, baked products, such as macarons [6]. It can also be stated that changes in hardness can be associated with the competition between the erythritol and the whey proteins. Due to the high solubility of whey proteins, the available water for erythritol solubility is less, and the crystallization phenomenon was produced when erythritol was exposed to heat [53]. Jang et al. [60] also highlighted the molecular size of erythritol and its effect on decreasing water activity in comparison to sucrose. Since it is higher, it produced greater osmotic pressure than the identical solution containing sucrose. Additionally, the decreasing values of water activity and hardness observed for the macarons could be associated with a lower ability of erythritol to crystallize, together with its greater hygroscopicity. It is also worth noting that almond flour, as a component consisting mostly of fat, is a highly hydrophobic ingredient that can impair the interactions between the whey proteins and the formation of gel networks during baking. Moreover, the almond flour can promote the lubricating effect in fat and, as a consequence, produce macarons that are more susceptible to mechanical deformation and breaking. Based on the baking volume of the final products and their microstructure, it can be stated that, with the addition of erythritol, the amount of air bubbles increased, resulting in an increase in their density, affecting a decrease in hardness in the obtained macarons [55], which can be correlated with the rheological properties of the macaron batters before thermal solidification. Due to the similarities of sucrose and erythritol, it can be stated that the structure and texture of macarons are controlled by erythritol and its concentration via affecting batter rheology during solidification, phase transitions, and the thermosetting of whey proteins, as well as moisture distribution between the ingredients. Erythritol can also act as a plasticizer and a humectant in relation to final macaron structuring. Our results clearly demonstrate the validity of replacing sucrose with polyols of a comparable molecular weight [61,62].

The water activity values show that no microbial growth will occur, and the macarons are absolutely safe as the a_w_ for all samples is far below 0.5. Nascimento et al. [63] stated that a_w_ values between 0.25 and 0.50 for confectionery products are recognized as microbiologically stable products. For all the macarons analyzed over the entire range of tested erythritol, a_w_ exhibited a decrement with larger erythritol concentrations. The highest a_w_ value was recorded for the reference sample without erythritol E0 (0.207 ± 0.001) and the lowest for the macaron sample with the largest erythritol addition E60 (0.176 ± 0.001). Similar to sugars, erythritol is recognized as having a great affinity for water. As a result, erythritol binds the water via hydrogen bonds. This interaction resulted in a decrease in the water activity of the macarons. The decreased water activity contributes to shelf-life prolongation since bound water is inaccessible to bacterial growth [64]. In addition, the effect of extending the shelf life of macarons is compounded by WPI itself, for which the water activity was found to be 0.241 [10]. 

### 3.5. Microstructure of Macarons

Figure 3 shows the optical microscope images of the macaron surfaces. They can be a very useful model for the analysis of the conversion of batters into actual macarons. Various erythritol concentrations led to different variations in the microstructures of the samples. The images supported the earlier findings from the profilometer analysis, in which the surface roughness of the macarons decreased with increasing erythritol addition. It could be observed that the largest erythritol addition (E60) was characterized by amorphous regions, distinct holes, and a smaller number of large air pockets, which were most probably broken during the batter-baking process. Its microstructure showed an even distribution of small and medium bubbles that were evenly distributed throughout the macaron matrix and thicker lamellar walls compared to the samples without the erythritol addition (E0). This implied that the E40 erythritol addition was optimal for maintaining air bubbles in the obtained macarons, which was additionally supported by the improved rheological properties and stability of the corresponding batter. It could be assumed that the largest content of erythritol caused an increase in the viscosity of the batter, providing the thickest lamellar walls that were an effective barrier against protein adsorption and bubble formation at the interface. Raikos et al. [65] emphasized that larger saccharose concentrations modified the air bubble size and thickness of lamellar walls. Berry et al. [52] claimed that saccharose (as a bulking agent) determined the microstructure of the sponge cake obtained from a whey protein isolate. Yang and Foegeding [49] found that the addition of up to 10% sugar was more effective in holding a WPI-based meringue matrix. Based on the analyses, it could be postulated that analogous mechanisms were also observed for erythritol due to its chemical similarity to sucrose.

Figure 4 presents the actual photographs of the macarons and their side profiles. It is worth noting that they can be associated with the rheological properties of the samples as they occupied a greater area at their base. The top views proved the observations made earlier in that the surface roughness of the macarons decreased with increasing erythritol concentration. The smoothing effect was very visible, even to the naked eye. The increased erythritol addition led to an increased final volume in the macarons. The smallest macaron volume for the control sample (E0) could be attributed to the collapse of the foam matrix during baking, which indicated very clearly that the bulking of the batter containing just WPI alone was insufficient. The macaron sample containing the highest concentration of erythritol (E60) exhibited the largest volume despite the fact that a decrease in the rheological properties and stability was found for the corresponding batter. According to Raikos et al. [65], these relationships are very difficult to correlate for such aerated systems because air bubbles in the batters can expand and/or coalesce during their solidification. Manisha et al. [66] emphasized the significance of the water evaporation effect related to macaron batter setting, meaning their volumes can vary. Regarding the volume of the properties of the macarons, they followed a trend observed in other studies [67], in which the lower the specific volume, the greater the macaron hardness.

## 4. Conclusions 

For the batters, the yield stress, storage, and loss moduli increased with a content of up to 40 g of erythritol and then decreased at 60 g. The E40 erythritol addition offered the greatest stability to the macaron batter, which was confirmed by the analyses in the Turbiscan and corresponding TSI values. The E60 erythritol content was detrimental to the rheological properties of the batter and its stability before baking; however, it led to an improvement in the final volume of the macarons. The increased erythritol concentration resulted in a smoother surface on the obtained macarons over the entire range of tested erythritol concentrations and demonstrated a major impact on the color parameters. With larger levels of erythritol, the texture characteristics, such as hardness and fracturability, decreased. The optical microscopy allowed us to explain variations in their textural and rheological properties and the stabilities of the corresponding batters. A higher erythritol content led to increased water activity values and extensions to the shelf life of the tested macarons, indicating a protective effect of the whey protein isolate and erythritol against microbial spoilage. The resulting macarons complement the rather limited catalog of high-protein functional foods, like yogurts, cheeses, and protein shakes, and for which, unlike macarons, do not require refrigeration temperatures. 

Based on the following research, it can be stated that whey protein isolate and erythritol can be utilized by the food industry for the production of high-protein, sugar-free macarons. Erythritol added in the proposed quantities allowed for the elimination of saccharose completely in the production of macarons. These compounds could successfully provide reproducible results, with increasing consistency and repeatability in confectioneries as well as home kitchens using very common equipment and basic ingredients. Moreover, they both improved the nutritional value and reduced the caloricity of the macarons. However, it is worth mentioning that the manufacturing costs will be higher in comparison to the traditional macaron recipe, but in this case, the consumer health issues related to excessive sugar consumption should be given top priority. Finally, this novel approach is aimed at consumers such as sportsmen and physically active people who are willing to cover extra costs for special functional foods that meet their increased wholesome protein requirements.

## Figures and Tables

**Figure 1 foods-12-01547-f001:**
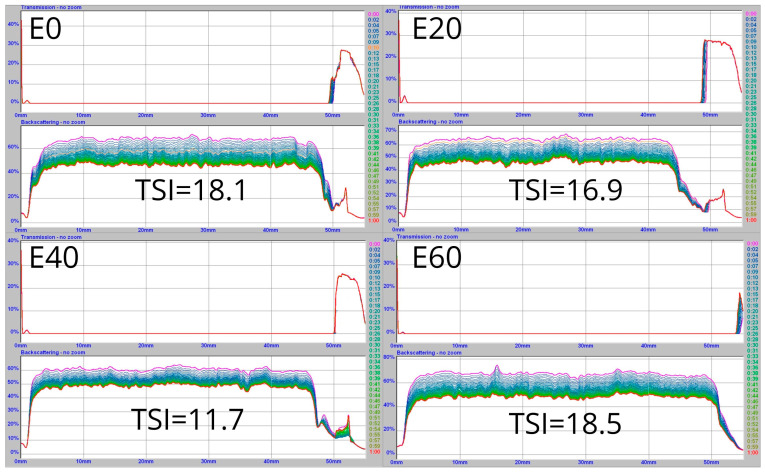
Transmission and backscattering and corresponding Turbiscan stability index (TSI) values of macaron batters before baking.

**Figure 2 foods-12-01547-f002:**
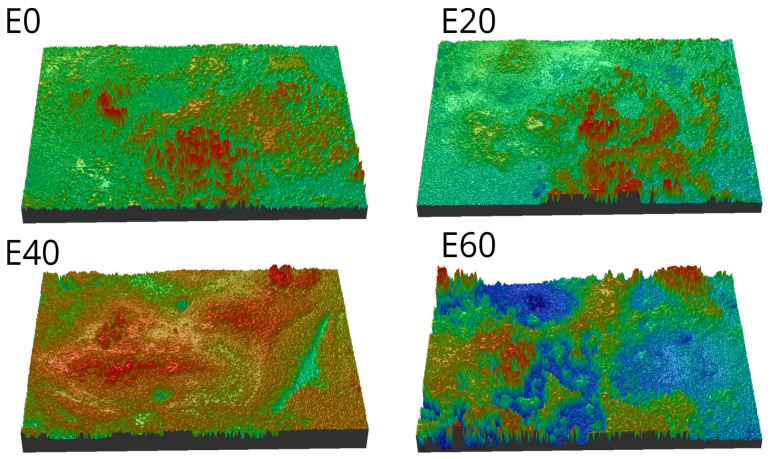
Optical profilometer images (surface 0.9 × 1.3 mm) of the obtained macaron samples. The largest roughness values are represented by the red regions and the smallest by the blue ones.

**Figure 3 foods-12-01547-f003:**
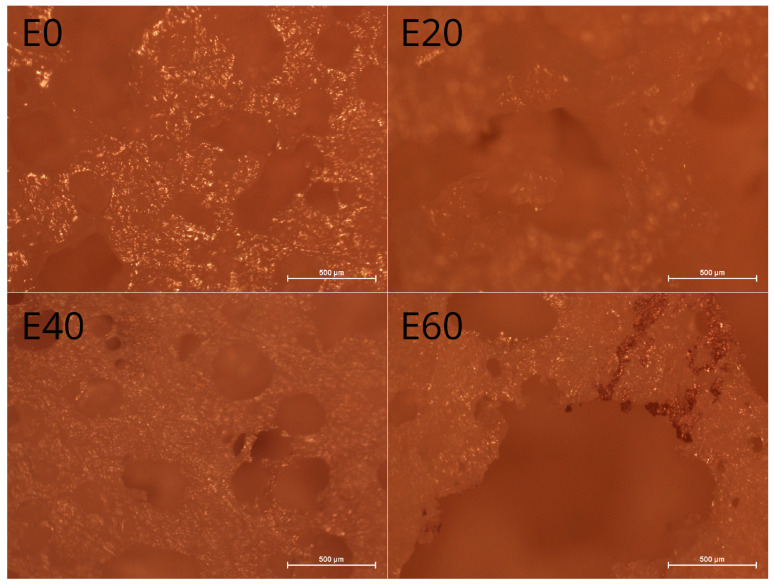
The optical microscope images (magnification 40×) for the obtained macarons.

**Figure 4 foods-12-01547-f004:**
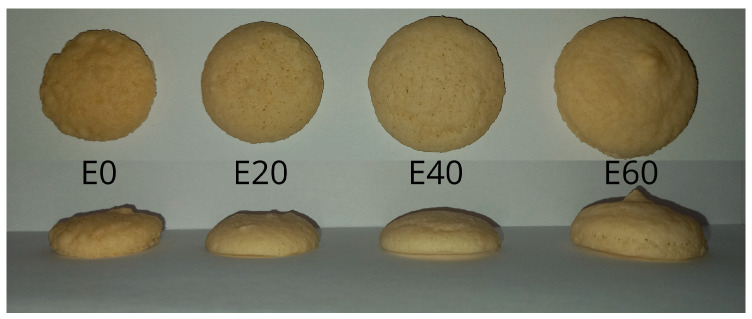
Images of the top views and side profiles of the obtained macarons.

**Table 1 foods-12-01547-t001:** The sample codes and composition of the obtained macarons.

Sample	Macaron Compositions
E0	20% WPI dispersion/0 g erythritol/125 g almond flour
E20	20% WPI dispersion/20 g erythritol/125 g almond flour
E40	20% WPI dispersion/40 g erythritol/125 g almond flour
E60	20% WPI dispersion/60 g erythritol/125 g almond flour

**Table 2 foods-12-01547-t002:** The effect of erythritol on the yield stress (Pa), storage modulus (Pa), loss modulus (Pa), and phase angle (°) values of the macaron batters before solidification.

Sample	τ (Pa)	G′ (Pa)	G″ (Pa)	δ (°)
E0	27.10 ^c^ ± 1.2	1028.33 ^c^ ± 12.02	263.18 ^b^ ± 5.70	14.36 ^b^ ± 0.12
E20	34.40 ^b^ ± 1.7	1379.93 ^b^ ± 7.01	343.56 ^a^ ± 6.13	14.05 ^c^ ± 0.05
E40	40.75 ^a^ ± 2.1	1399.35 ^a^ ± 10.86	347.61 ^a^ ± 5.33	13.98 ^d^ ± 0.02
E60	21.30 ^d^ ± 1.1	716.94 ^d^ ± 9.36	192.62 ^c^ ± 6.44	15.04 ^a^ ± 0.10

The differences among the mean values in the column designated with different letters are statistically significant (*p* < 0.05).

**Table 3 foods-12-01547-t003:** The surface roughness parameters and the color co-ordinates of the obtained macarons. R_a_—the average roughness, R_q_—the quadratic mean of surface roughness, R_t_—the maximum roughness height, L*—the brightness, a*: ± red-green, and b*: yellow-blue.

Sample	R_a_ (μm)	R_q_ (μm)	R_t_ (μm)	L*	a*	b*
E0	37.51 ^a^ ± 2.49	45.75 ^a^ ± 1.25	473.53 ^a^ ± 4.21	80.0 ^d^ ± 0.00	4.28 ^a^ ± 0.48	16.71 ^a^ ± 0.48
E20	26.77 ^b^ ± 1.17	36.79 ^b^ ± 1.01	416.13 ^b^ ± 7.27	82.00 ^c^ ±0.00	3.57 ^ab^ ± 0.53	13.00 ^b^ ± 0.57
E40	22.72 ^c^ ± 1.24	33.13 ^c^ ± 0.92	317.17 ^c^ ± 6.32	83.28 ^b^ ± 0.48	3.00 ^b^ ± 0.00	10.71 ^c^ ± 0.57
E60	17.71 ^d^ ± 2.97	25.21 ^d^ ± 5.35	283.35 ^d^ ± 13.25	84.00 ^a^ ± 0.00	2.85 ^b^ ± 0.37	10.00 ^d^ ± 0.00

The differences among the mean values in the column (designated with different letters) are statistically significant (*p* < 0.05).

**Table 4 foods-12-01547-t004:** The effect of erythritol on the textural (hardness and fracturability) parameters and water activity of the obtained macarons.

Sample	Hardness (N)	Fracturability (N)	Water Activity (a_w_)
E0	63.81 ^a^ ± 2.24	63.08 ^a^ ± 3.02	0.207 ^a^ ± 0.001
E20	19.95 ^b^ ± 0.16	18.33 ^b^ ± 0.86	0.204 ^b^ ± 0.001
E40	16.11 ^c^ ± 1.56	15.34 ^c^ ± 0.75	0.196 ^c^ ± 0.001
E60	11.84 ^d^ ± 0.19	10.98 ^d^ ± 1.14	0.176 ^d^ ± 0.001

The differences among the mean values in the column designated with different letters are statistically significant (*p* < 0.05).

## Data Availability

The data presented in this study are available on request from the corresponding author.

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
