# Peer review of "The Effect of Erythritol on the Physicochemical Properties of Reformulated, High-Protein, and Sugar-Free Macarons Produced from Whey Protein Isolate Intended for Diabetics, Athletes, and Physically Active People"

_foods, 2023, doi:10.3390/foods12071547_

Round 1
Reviewer 1 Report
The article "The Effect of Erythritol on Physylumical Properties of Reformulated, High-Protein, Sugar-Free Macarons Produced from Whey Protein Isolalate Intended for Diabetic, Athletes and Physically Active People" has been reviewed. The article is considered to be well developed and meets the requirements of a high -level publication.
I consider that in the introduction some relevant data on the overweight must be offered and its incidence in certain sectors of the society that most consume these foods.
Section 2.4 Explain why it is important to know the rheology of the prepared mass
Section 2.5. Explain why Batter Stability is important
The water activity is affected by the temperature, then please specify to what conditions the food pieces were evaluated.
The figures and tables are of interest, however I would like to see clearly an argument with the results obtained by other researchers and observe a coincidence or not of the results. especially the physical effects that can be observed with the effect of the concentration of the sugar substitute.
The conclusions lack a little order to establish ideas.
Small comments were placed in the attached file.
I want to congratulate researchers for the quality and presentation of work

Author Response
April 1st, 2023
Dear Prof. Jurlin Zhao and Reviewers
Our manuscript foods-2297048 transferred from International Journal of Environmental Research and Public Health entitled "The Effect of Erythritol on Physicochemical Properties of Reformulated, High-Protein, Sugar-Free Macarons Produced from Whey Protein Isolate Intended for Diabetics, Athletes and Physically Active People” has been revised and is being re-submitted for publication in Foods. We have carefully considered each of the comments and made the appropriate revisions in the manuscript. All major changes in the text are marked in green. An itemized list of our responses to each of the comments is included below.
Thank you for your kind attention.
Yours faithfully,
Maciej Nastaj et al.
Editor # foods-2297048
Thank you kindly for the entire reviewing process. I hope all of the amends implemented will indeed boost the robustness and impact of our paper. On behalf of myself and co-authors I would like to thank for agreeing to the article transfer from from International Journal of Environmental Research and Public Health to Foods. We would additionally like to emphasize that the text underwent professional English proofreading by a native speaker.
Thank you very much.
Kind regards, Maciej Nastaj et al.
Reviewer 1 #foods-2297048
The article "The Effect of Erythritol on Physicochemical Properties of Reformulated, High-Protein, Sugar-Free Macarons Produced from Whey Protein Isolate Intended for Diabetic, Athletes and Physically Active People" has been reviewed. The article is considered to be well developed and meets the requirements of a high -level publication.
I consider that in the introduction some relevant data on the overweight must be offered and its incidence in certain sectors of the society that most consume these foods.
Thank you kindly for your comment. We absolutely agree, the issue of obesity is missing and in the context of civilization diseases is very important. The introduction section has been supplemented with relevant content from the latest literature.
Renzetti, S.; van der Sman, R.G.M. Food texture design in sugar reduced cakes: Predicting batters rheology andphysical properties of cakes from physicochemical principles. Food Hydrocoll. 2022, 131, 107795
Bordier, V.; Teysseire, F.; Schlotterbeck, G.; Senner F.; Beglinger, C.; Meyer-Gerspach, A.C.; Wölnerhanssen, B.K. Effect of a Chronic Intake of the Natural Sweeteners Xylitol and Erythritol on Glucose Absorption in Humans with Obesity. Nutrients 2021, 13, 3950.
Section 2.4 Explain why it is important to know the rheology of the prepared mass
Thank you kindly for your comment. This section now explains the validity of rheological analyses of the batters. The following sentence has been added: ”The rheology of the batters relates to their stability and determines further surfacial, textural, structural properties of macarons obtained”.
Section 2.5. Explain why Batter Stability is important
Thank you kindly for your comment. The following sentence has been added: “In addition to the rheological properties of the batters, assessment of their stabilities is also of significant importance. Analysis in Turbiscan allows to monitor all instability mechanisms (gravitational drainage, coalescence, and disproportionation) that occur simultaneously in the batters. This will facilitate to control the quality of the finished product through appropriate selection of ingredients and elimination of the defects”.
The water activity is affected by the temperature, then please specify to what conditions the food pieces were evaluated.
Thank you kindly for your comment. Yes, of course, water activity is temperature dependent. We mention in this paragraph that the measurement was made at 25 degrees. And the apparatus was calibrated for this dedicated temperature with solutions of various salts according to the recommendations of the manufacturer of the device. This specific temperature was also suggested by LabMaster-aw manufacturer for proper storing of the resulting macarons and other products of a similar type.
The figures and tables are of interest, however I would like to see clearly an argument with the results obtained by other researchers and observe a coincidence or not of the results. especially the physical effects that can be observed with the effect of the concentration of the sugar substitute.
Thank you kindly for your comment. The literature in this area has been supplemented based on similar studies. Despite the extensive literature on sugar functionality in bakery, the topic of macarons is basically non-existent. On the other hand, we are happy to be pioneers. Currently, our research team is obtaining and a model high-protein macaroons with reduced sugar content. Our results will be published soon, which will fill the existing gap and give a wide scope for comparison.
I think it is also worth highlighting the similarities in the chemical structure of sucrose and erythritol, which we do several times in the text. As recommended, we supplemented the literature data on sugar by quoting the following publications.
Renzetti, S.; van der Sman, R.G.M. Food texture design in sugar reduced cakes: Predicting batters rheology andphysical properties of cakes from physicochemical principles. Food Hydrocoll. 2022, 131, 107795.
Slade, L., Kweon, M.; Levine, H. Exploration of the functionality of sugars in cake-baking, and effects on cake quality. Crit Rev Food Sci 2020, 61(2), 283–311.
Kweon, M.; Slade, L.; Levine, H. Cake baking with alternative carbohydrates for potential sucrose replacement. I. Functionality of small sugars and their effects on high-ratio cake-baking performance. Cereal Chem 2016, 93(6), 562–567.
Lau, C.K.; Dickinson, E. Instability and structural change in an aerated system containing egg albumen and invert sugar. Food Hydrocoll. 2005, 19, 111–121.
Belyakova, L.E.; Antipova, A.S.; Semenova, M.G.; Dickinson, E.; Merino, L.; Tsapkina, E.N. Effect of sucrose on molecular andinteraction parameters of sodium caseinate in aqueous solution: Relationship to protein gelation. Colloids Surf. B 2003, 31, 31–46.
Dickinson, E.; Matia-Merino, L. Effect of sugars on the rheological properties of acid caseinate-stabilized emulsion gels. Food Hydrocoll. 2002, 16, 321–331.
The conclusions lack a little order to establish ideas.
Thank you kindly for your comment. The order of sections has been swapped, from the specific derived from our research to the general ones. Now it actually makes more sense.
Small comments were placed in the attached file.
Thank you kindly for your comments. Responses to all comments are now included in resubmitted version of the manuscript.
I want to congratulate researchers for the quality and presentation of work
Thank you very much! On behalf of all the co-authors and myself, thank you for all your kind comments on our article. We tried very hard and are very happy that our work was highly evaluated. We appreciate that a lot!
Thank you very much. We would additionally like to emphasize that the text underwent professional English proofreading by a native speaker.
Kind regards, Maciej Nastaj et al.
Reviewer 2 Report
Dear Authors, the article is interesting and well written, being an interest topic for the actual context of food security.
The template is not of Foods Journal.
This research studies if the high-protein, sugar-free, WPI-based macarons can be produced with acceptable properties for diabetics, athletes and physically active people with increased protein requirements and reduced caloric value at the same time. Also, provides information on the effect of the composition of macarons with the addition of WPI and erythritol as these can be new potential compounds in the confectionery industry to produce novel functional foods.
Anyway considering also the field of research of this journal the work is very valuable and I would like to congratulate the authors for their work.
The aim of the study is very well defined.
The introduction section is very well documented and written.
Material and Methods
Line 119: Delete „protein”.
Lime 197: Delete „dose”.
Table 2: The parameters have to be defined, even on title, or under table.
The experimental part is accurately done and very complex, and the results are well-presented bringing solid discussions.
Also, the Conclusion Section is very well summarised and highlights the results of the study.
Author Response
April 1st, 2023
Dear Prof. Jurlin Zhao and Reviewers
Our manuscript foods-2297048 transferred from International Journal of Environmental Research and Public Health entitled "The Effect of Erythritol on Physicochemical Properties of Reformulated, High-Protein, Sugar-Free Macarons Produced from Whey Protein Isolate Intended for Diabetics, Athletes and Physically Active People” has been revised and is being re-submitted for publication in Foods. We have carefully considered each of the comments and made the appropriate revisions in the manuscript. All major changes in the text are marked in green. An itemized list of our responses to each of the comments is included below.
Thank you for your kind attention.
Yours faithfully,
Maciej Nastaj et al.
Editor # foods-2297048
Thank you kindly for the entire reviewing process. I hope all of the amends implemented will indeed boost the robustness and impact of our paper. On behalf of myself and co-authors I would like to thank for agreeing to the article transfer from from International Journal of Environmental Research and Public Health to Foods. We would additionally like to emphasize that the text underwent professional English proofreading by a native speaker.
Thank you very much.
Kind regards, Maciej Nastaj et al.
Reviewer 2 #foods-2297048
Dear Authors, the article is interesting and well written, being an interest topic for the actual context of food security
Dear Reviewer,
On behalf of all the co-authors and myself, thank you very much for all your kind comments on our article. We tried very hard and are very happy that our work was highly evaluated. We appreciate that a lot!
The template is not of Foods Journal.
Thank you kindly for your comment. This has already been fixed and appropriate template has been used. Initially, our article was sent to International Journal of Environmental Research and Public Health, despite receiving positive reviews, the lead editor suggested transfer to Foods.
This research studies if the high-protein, sugar-free, WPI-based macarons can be produced with acceptable properties for diabetics, athletes and physically active people with increased protein requirements and reduced caloric value at the same time. Also, provides information on the effect of the composition of macarons with the addition of WPI and erythritol as these can be new potential compounds in the confectionery industry to produce novel functional foods.
Anyway considering also the field of research of this journal the work is very valuable and I would like to congratulate the authors for their work.
The aim of the study is very well defined.
The introduction section is very well documented and written.
Thank you kindly for your comments.
Material and Methods
Line 119: Delete „protein”
Thank you kindly for your comment. The extra “protein” word has been deleted.
Line 137: Delete „dose”
Thank you kindly for your comment. The extra “dose” word has been deleted.
Table 2: The parameters have to be defined, even on title, or under table.
I think this comment was about table 4, in the attached pdf there is a note marked in yellow in the text. The textural parameters (hardness and fracturability) have been defined on the table title.
The experimental part is accurately done and very complex, and the results are well-presented bringing solid discussions.
Also, the Conclusion Section is very well summarised and highlights the results of the study.
Thank you kindly for your comments.
Thank you very much. We would additionally like to emphasize that the text underwent professional English proofreading by a native speaker.
Kind regards, Maciej Nastaj et al.